# Hepatic macrophages play critical roles in the establishment and growth of hydatid cysts in the liver during *Echinococcus granulosus* sensu stricto infection

Hui Wang[1,2,3☯*], Qian Yu[2☯], Mingkun Wang[2☯], Jiao Hou[1], Maolin Wang[1,4], Xuejiao Kang[2], Xinling Hou[2], Dewei Li[2], Zibigu Rousu[2], Tiemin Jiang[1,4], Jing Li[2], Hao Wen[1,3*], Chuanshan Zhang[1,2,3*]

**1** State Key Laboratory of Pathogenesis, Prevention and Treatment of High Incidence Diseases in Central Asia, Clinical Medicine Institute, The First Affiliated Hospital of Xinjiang Medical University, Urumqi, Xinjiang, China, **2** Basic Medical College, Xinjiang Medical University, Urumqi, Xinjiang, China, **3** Xinjiang Key Laboratory of Echinococcosis, Clinical Medicine Institute, World Health Organization Collaborating Centre on Prevention and Case Management of Echinococcosis, The First Affiliated Hospital of Xinjiang Medical University, Urumqi, Xinjiang, China, **4** Department of Hepatic Hydatid and Hepatobiliary Surgery, Digestive and Vascular Surgery Centre, The First Affiliated Hospital of Xinjiang Medical University, Urumqi, Xinjiang, China

☯ These authors contributed equally to this work.

* wangh0923@126.com (HWA); dr.wenhao@163.com (HWE); dashan0518@126.com (CZ)

**Data Availability Statement:** All data are in the manuscript and/or supporting information files.

## Abstract

Cystic echinococcosis (CE) is a worldwide neglected zoonotic disease caused by infection with the larval stage of the tapeworm *Echinococcus granulosus* sensu lato *(E. granulosus s.l.)*, which predominantly resides in the liver accompanied by mild inflammation. Macrophages constitute the main cellular component of the liver and play a central role in controlling the progression of inflammation and liver fibrosis. However, the role of hepatic macrophages in the establishment and growth of hydatid cysts in the liver during *E. granulosus* sensu stricto (*E. granulosus s.s.*) infection has not been fully elucidated. Here, we showed that CD68[+] macrophages accumulated in pericystic areas of the liver and that the expression of CD163, a marker of anti-inflammatory macrophages, was more evident in active CE patients than in inactive CE patients. Moreover, in a mouse model of *E. granulosus s.s.* infection, the pool of hepatic macrophages expanded dramatically through the attraction of massive amounts of monocyte-derived macrophages (MoMFs) to the infection site. These infiltrating macrophages preferentially polarized toward an iNOS[+] proinflammatory phenotype at the early stage and then toward a CD206[+] anti-inflammatory phenotype at the late stage. Notably, the resident Kupffer cells (KCs) predominantly maintained an anti-inflammatory phenotype to favor persistent *E. granulosus s.s.* infection. In addition, depletion of hepatic macrophages promoted *E. granulosus s.s.* larval establishment and growth partially by inhibiting CD4[+] T-cell recruitment and liver fibrosis. The above findings demonstrated that hepatic macrophages play a vital role in the progression of CE, contributing to a better understanding of the local inflammatory responses

**Funding:** This research was supported by the National Key Research and Development Program of China (2021YFC2300800, 2021YFC2300801 to CZ, 2021YFC2300802 to HW), the Xinjiang Uygur Autonomous Region Tianshan Innovation Team Program (2023D14009 to HW), the Distinguished Young Scholars Program of Xinjiang Uygur Autonomous Region (2020Q007 to HW), National Natural Science Foundation of China (82160397, 81860359 and 81660342 to HW), and the State Key Laboratory of Pathogenesis, Prevention and Treatment of Central Asia High Incidence Diseases Fund (SKL-HIDCA-2022-BC1 to HW).The funders had no role in study design, data collection and analysis, decision to publish, or preparation of the manuscript.

**Competing interests:** The authors have declared that no competing interests exist.

surrounding hydatid cysts and possibly facilitating the design of novel therapeutic approaches for CE.

## Author summary

Cystic echinococcosis (CE) is one of the neglected zoonoses caused by *Echinococcus granulosus s.l.* larvae and is a worldwide public health problem. Previous study showed that CE is usually characterized by mild local inflammation, indicating that *E. granulosus s.l.* can control liver inflammation to shape an 'immunosuppressive' microenvironment, a balance between pro- and anti-inflammatory responses to maintain immune homoeostasis during persistent infection. Here, by combining data from human CE samples and animal models, we found that hepatic macrophages are involved in the control of liver inflammation to favor persistent *E. granulosus s.s.* infection. The pool of hepatic macrophages expanded dramatically at the early stage through the attraction of massive amounts of MoMFs that preferentially initially polarized toward a proinflammatory phenotype and then toward an anti-inflammatory phenotype to favor persistent infection at the late stage after infection. Depletion of hepatic macrophages accelerated hydatid cyst establishment and growth by inhibiting CD4$^+$ T-cell infiltration and liver fibrosis during *E. granulosus s.s.* infection. Our findings contribute to a better understanding of the local microenvironment in CE and possibly facilitate the design of novel therapeutic approaches for CE that target hepatic macrophages.

## Introduction

Cystic echinococcosis (CE) is a neglected zoonotic disease caused by infection with the larval stage of the tapeworm *Echinococcus granulosus* sensu stricto (*E. granulosus s.s.*), which forms hydatid cysts and has a high prevalence in western China, Central Asia, South America, Mediterranean countries and eastern Africa, affecting humans as well as livestock [1–3]. Hydatid cysts form predominantly in the liver (70%) and less frequently in the lungs (20%) and other organs (10%) [4]. Hydatid cysts grow similar to a unilocular bladder filled with cyst fluid and protoscoleces (PSCs). If a hydatid cyst ruptures within the human host, the released PSC can form a new cyst, which is also a life-threatening form of human CE [5]. Population screening has shown that hydatid cysts grow very slowly in the human liver and that their diameter can reach several centimeters, accompanied by mild inflammation [6, 7]. These properties indicate that *E. granulosus s.s.* larvae can control host inflammation to avoid clearance. Therefore, it is necessary to elucidate the underlying mechanism by which the parasite modulates the host immune response to favor its establishment and persistent infection in the host and to identify new potential targets for CE therapy.

Histological analysis showed that the hydatid cyst consists of an inner germinal layer (GL) and an outer laminated layer (LL) in the livers of CE patients; these structures are directly encapsulated by an adventitial layer formed by the host. The adventitial layer is surrounded by collagen fibers and periparasitic immune cells, including infiltrating T lymphocytes, granulocytes, and especially palisading monocytes and/or macrophages, which form an immune microenvironment surrounding the cyst that is favorable for persistent infection with *E. granulosus s.s.* larvae [7, 8]. In our prior work, we also found that T lymphocytes are involved in the formation of the liver immune microenvironment and are closely associated with cyst viability

and establishment in the liver in both CE patients and *E. granulosus s.s.*-infected mice [9–11]. Macrophages are the first line of host defense against infection and one of the predominant immune cells at the site of local responses to hydatid cysts in humans, mice and sheep [7, 8, 12, 13], indicating that macrophages may be critical for hydatid cyst establishment. However, the role of macrophages in the liver during the establishment and growth of *E. granulosus s.s.* larvae has not yet been fully elucidated.

Macrophages, which constitute the main cellular component in the liver, are a highly heterogeneous population [14]. The hepatic macrophage population consists of liver-resident Kupffer cells (KCs), which constitute the largest population of tissue macrophages in human organs, and recruited monocyte-derived macrophages (MoMFs), which rapidly expand during liver injury [15]. However, little is known about the arrangement of liver-resident and recruited macrophages around hydatid cysts in CE patients and *E. granulosus s.s.*-infected mouse models. Moreover, macrophages are plastic cells capable of polarizing toward the proinflammatory M1 phenotype or the anti-inflammatory M2 phenotype in different microenvironments or under exposure to different stimuli [16]. Our previous studies also showed that large amounts of macrophages were recruited into the peritoneal cavity and preferentially polarized toward the M2 phenotype in a mouse model established by intraperitoneal (i.p.) inoculation with viable *E. granulosus s.s.* larvae [17, 18]. Currently, little information is available on the effects of *E. granulosus s.s.* larval infection on hepatic macrophage phenotypes and functions in the liver immune microenvironment.

In the present study, we first identified the characteristics of macrophage infiltration surrounding pericysts in the livers of CE patients with different degrees of cyst viability and then analyzed the hepatic macrophage composition and phenotype in a mouse model of *E. granulosus s.s.* infection at different times. Furthermore, we demonstrated that depletion of hepatic macrophages using clodronate liposomes (CL) impaired the ability to clear *E. granulosus s.s.* larvae and promoted *E. granulosus s.s.* establishment and growth partially by inhibiting liver fibrosis and CD4+ T-cell recruitment in the mouse model.

## Materials and methods

### Ethics statement

The involvement of human participants in this study was approved by the Ethics Committee of the First Affiliated Hospital of Xinjiang Medical University (Approval No. S20130418-3) and written informed consent was obtained from each participant or the guardian.

Eight-week-old female C57BL/6 wild-type (WT) mice were purchased from Beijing Vital River Experimental Animal Technology Co., Ltd. and housed in a specific pathogen-free environment at the Animal Facility of Xinjiang Medical University. All mice received humane care, and all experimental protocols involving mice were carried out in accordance with the Guide for the Care and Use of Laboratory Animals and approved by the Ethics Committee of the First Affiliated Hospital of Xinjiang Medical University (Approval No. 20170809–01).

### CE patients and sample collection

The inclusion criteria for the CE patients enrolled in this study were as follows: (i) patients who were diagnosed by clinical symptoms or ultrasonography (US) alone or in combination with computed tomography (CT) and with the diagnosis confirmed via liver biopsy; (ii) patients who had cysts only in the liver, had undergone liver resection or total cystectomy, and had no previous history of treatment with antiparasitic agents; and (iii) patients without any other infectious comorbidities, including autoimmune diseases or viral hepatitis. In addition, according to the World Health Organization Informal Working Group on Echinococcosis

(WHO-IWGE) classification system [19], hepatic CE cysts are classified into five types (CE1 to CE5), which are divided into three stages: active CE (CE1 and CE2), transitional CE (CE3) and inactive CE (CE4 and CE5) [20]. CE3 cysts are at a transitional stage, with an equal probability of being viable or nonviable. Thus, patients who had CE3 cysts were not included in this study. The demographical and clinical characteristics of the CE patients are summarized in S1 Table. Samples of nearby liver tissues containing the cyst lumen, fibrous capsule and liver parenchyma adjacent to the hydatid cyst lesion were collected from each patient who underwent liver resection, as we previously reported [10, 21]. All collected specimens were fixed with 4% (v/v) paraformaldehyde for 48–72 hours (hr) and then embedded in paraffin for pathological and immunohistochemical (IHC) analysis.

### Hepatic mouse model of *E. granulosus s.s.* infection

*E. granulosus s.s.* PSCs were obtained under aseptic conditions from liver hydatid cysts in fertile sheep [22]. PSC viability was determined by staining with 0.1% methylene blue, and only PSC samples with greater than 95% viability were used for mouse infection. The hepatic mouse model of *E. granulosus s.s.* was established as previously described [23]. In brief, each mouse in the low-dose group (LD), medium-dose group (MD) and high-dose group (HD) was inoculated via the hepatic portal vein with 50, 500 and 2000 PSCs in saline, respectively. Control group (Con) mice were injected with the same volume of saline alone.

### Depletion of hepatic macrophages *in vivo*

Based on our prior study [21], mice were injected intraperitoneally (i.p.) with CL (ClodronateLiposomes.org, Amsterdam, Netherlands; 100 μL of suspension/10 grams of animal weight) to deplete hepatic macrophages. CL were administered every week to maintain depletion. Control mice were injected with the same volume of empty liposome-PBS (PL). For early hepatic macrophage depletion, mice were administered (i.p.) CL or PL on day 3 before inoculation of *E. granulosus* s.s. PSCs via the hepatic portal vein and again on day 3 and day 10 after PSC inoculation. For late hepatic macrophage depletion, mice were first infected with *E. granulosus* s.s. PSCs via the hepatic portal vein and were then administered (i.p.) CL or PL once a week for 6 weeks beginning 18 weeks after PSC inoculation. Four to five mice per group (CL and PL) were sacrificed at each depletion stage (early stage: 1 and 2 weeks; late stage: 24 weeks) after PSC inoculation to evaluate the liver parasite burden and histopathological changes.

### Histopathological analysis

Paraffin-embedded blocks of liver tissue from CE patients were sliced into 4-μm-thick sections, which were used to visualize inflammatory cell infiltration and changes in general histology by hematoxylin-eosin (H&E) staining and to visualize hepatic collagen deposition for fibrosis assessment by Sirius red (SR) or Masson staining. The widths (μm) of the inflammatory halo and the fibrous layer were quantified using cellSens Dimension software.

For histopathological analysis of liver samples from the *E. granulosus s.s.* infection mouse model, 4–6 mice in the HD group were sacrificed at each infection stage (early stage: 2, 5, 8, 11 days and 2 weeks; middle stage: 12 weeks; late stage: 24 weeks) to study the dynamic pathological changes in hepatic tissue after infection. The whole liver surface was carefully screened for hepatic lesions for preliminary assessment of the intensity of *E. granulosus s.s.* infection, and the livers were then collected and divided into three sections, which were fixed with 4% (v/v) paraformaldehyde for 48 hr, embedded in paraffin and made into three FFPE blocks for H&E

staining. The numbers of intact PSCs in the liver in the CL and PL groups at 1 and 2 weeks were determined from the three sections.

## Immunohistochemical analysis

For IHC analysis, liver sections were prepared as previously described [21]. Sections were incubated at 4°C overnight with the following primary antibodies: anti-human CD68, 1:100, ab955; anti-human CD163, 1:500, ab182422; anti-human S100A9, 1:1000, ab63818; anti-human/mouse $\alpha$-SMA, 1:500, ab 124964; anti-mouse F4/80, 1:100, ab6640; anti-mouse CD4, 1:1000, ab183685 (all from Abcam, Cambridge, UK); and anti-mouse Collagen Type III Alpha 1 (Col3$\alpha$1), 1:2000 (Cat: 2273, Proteintech, Illinois, USA). The next day, the sections were washed three times with Tris-buffered saline and 0.1% Tween (TBST) for 15 min and then incubated with a secondary antibody for 1 hr at room temperature. Visualization of staining was performed with a diaminobenzidine (DAB) substrate kit (ab64238, Abcam, Cambridge, UK) according to the manufacturer's instructions. Then, the stained sections were examined and photographed at 100×, 200× or 400× magnification (3–5 fields/section/sample) using a digital image acquisition system (Olympus, Tokyo, Japan). The area of intense positive staining was measured using cellSens Dimension software (Olympus), and the data are presented as the percentages of positive area per field.

## Flow cytometric analysis

For flow cytometric analysis, 5–6 mice per dose group (Con, LD, MD and HD) were sacrificed, and nonparenchymal liver cells (NPLCs) were isolated from each mouse liver to analyze the composition and phenotype of hepatic macrophages at 2, 12 and 24 weeks [24]. After cell counting, $1.0 \times 10^6$ NPLCs were preincubated with an anti-mouse CD16/CD32 antibody (Fc Block; Cat# 101302, BioLegend, San Diego, CA) for 20 min at 4°C to block nonspecific antibody binding. For cell surface flow cytometry, cells were stained with a mixture of surface antibodies (anti-CD3-FITC, Clone 17A2, Cat# 100204; anti-CD19-FITC, Clone 6D5, Cat# 115506; anti-NK1.1-FITC, Clone PK136, Cat# 108706; anti-CD45-PerCP/Cy5.5, Clone 30-F11, Cat# 103132; anti-Ly6G-APC/Cy7, Clone 1A8, Cat# 127624; anti-CD11b-Brilliant Violet 650, Clone M1/70, Cat# 101239; anti-F4/80-PE, Clone BM8, Cat# 123110; all from BioLegend, San Diego, CA) for 30 min at 4°C in the dark. For intracellular flow cytometry, cells were fixed, permeabilized, and stained with intracellular antibodies (anti-CD206-APC, Clone 15–2, Cat# 321110, BioLegend, San Diego, CA; anti-iNOS-PE/Cy7, Clone CXNFT, Cat# 25-5920-82, eBioscience, California, USA) according to the manufacturer's instructions (BD Biosciences; cat no. 554714). Approximately $2.0 \times 10^5$ cells from each sample were acquired on an LSRFortessa flow cytometer (BD Immunocytometry Systems, San Jose, CA, USA) for data analysis. The percentages of different hepatic macrophage subsets in the liver were determined using FlowJo software (version V10; Tree Star, Inc., Ashland, OR, USA). The absolute cell numbers of hepatic macrophage subsets were then calculated by the following formula: number of isolated NPLCs × percentage of hepatic macrophages among the NPLCs.

## Statistical analysis

All statistical analyses were performed using GraphPad Prism 8.0 software (GraphPad Software, San Diego, CA). After normality testing of the data sets, the appropriate parametric or nonparametric tests were used. For comparisons among three or more groups, ordinary one-way ANOVA and Tukey's multiple comparisons test were used if the data were normally distributed, and the Kruskal–Wallis test and Dunn's multiple comparisons test were used if the data were nonnormally distributed. For comparisons between two groups, Student's t test was

used for normally distributed data, and the Mann–Whitney U test was chosen for nonnormally distributed data. The statistical significance level was set at $P < 0.05$ ($P$ values are expressed as follows: $^*P < 0.05$; $^{**}P < 0.01$; $^{***}P < 0.001$ and $^{****}P < 0.0001$).

## Results

### Demographic and clinical characteristics of CE patients

A total of 41 CE patients were enrolled in the present study. The age (years) of the patients ranged from 4 to 68, with a mean age of 42. Twenty-three (56.1%) patients were female, and 18 (43.9%) patients were male. With regard to the stages of CE cyst activity, the cysts were classified as CE1 in 13 (31.7%) patients, as CE2 in 19 (46.3%) patients, and as CE4 in 9 (22.0%) patients. With regard to the number of CE cysts, 27 (65.9%) patients had one cyst, and 14 (34.1%) patients had multiple cysts. With regard to the location of CE cysts, the cysts were located in the right lobe of the liver in 31 (75.6%) patients, the left lobe in 6 (14.6%) patients, and in both the right and left lobes in 4 (9.6%) patients. The diameter of the cysts ranged from 1.3 to 17.2 cm, with a mean of 7.2 ± 3.5 cm. Furthermore, the levels (U/L) of serum ALT, AST and ALP, which reflects the liver function of CE patients, were higher in patients with active CE1, CE2 cysts than in patients with inactive CE4 cysts (ALT: CE1 vs. CE2 vs. CE4: 58.8 ± 13.7 vs. 46.8 ± 11.6 vs. 28.3 ± 8.0 U/L; AST: CE1 vs. CE2 vs. CE4: 48.8 ± 10.4 vs. 47.5± 17.5 vs. 28.3 ± 5.5 U/L; ALP: CE1 vs. CE2 vs. CE4: 113.2 ± 17.9 vs. 127.3 ± 24.4 vs. 79.9 ± 9.3 U/L) (S1 Table).

### Macrophages accumulated in the pericystic areas of the liver in CE patients with different cyst activity stages

Histopathological analysis of livers from CE patients showed a typical inflammatory microenvironment around hydatid cyst lesions, including a thick cell-free LL with a rather reddish appearance and prominent striation and a fibrotic rim, a peripheral halo of inflammatory exudate, and normal liver tissue (Fig 1A). The thickness (μm) of the aggregated inflammatory cell halo surrounding the cysts, which reflects the intensity of the inflammatory immune reaction, was greater in active CE2 cysts than in inactive CE4 cysts (CE1 vs. CE2 vs. CE4: 81.49 ± 8.36 vs. 115.3 ± 15.87 vs. 54.82 ± 3.53 μm, $P = 0.0151$) (Fig 1A and 1B). IHC analysis showed that CD68$^+$ macrophages, CD163$^+$ anti-inflammatory macrophages and S100A9$^+$ proinflammatory macrophages accumulated in the pericystic area (Fig 1D). The percentages of the CD68-positive area ($P = 0.0002$) and CD163-positive area ($P = 0.0449$) were significantly higher in patients with active cysts than in patients with inactive cysts (Fig 1E and 1F). The percentage of the S100A9-positive area did not differ significantly among the groups of CE patients with cysts at different stages, but the percentage of the S100A9-positive area was significantly higher than that of the CD163-positive area in the inactive CE4 specimens (Fig 1G and 1H). In addition, the size of the fibrotic area surrounding the cysts, which was evaluated by the width of the fibrous layer (μm) and by α-SMA, SR and Masson staining, did not differ significantly among the groups of CE patients with cysts at different stage (Figs 1C and S1).

### Macrophages accumulated in pericystic areas of the liver in the mouse model during *E. granulosus s.s.* larval establishment

To investigate the dynamic changes in macrophage accumulation in the pericystic areas of the liver during the course of infection, we established a hepatic experimental mouse model of *E. granulosus s.s.* via hepatic portal vein inoculation. The macroscopic views of the lesions at the surface of the infected livers showed areas of necrosis (diffuse white patchy and congestive

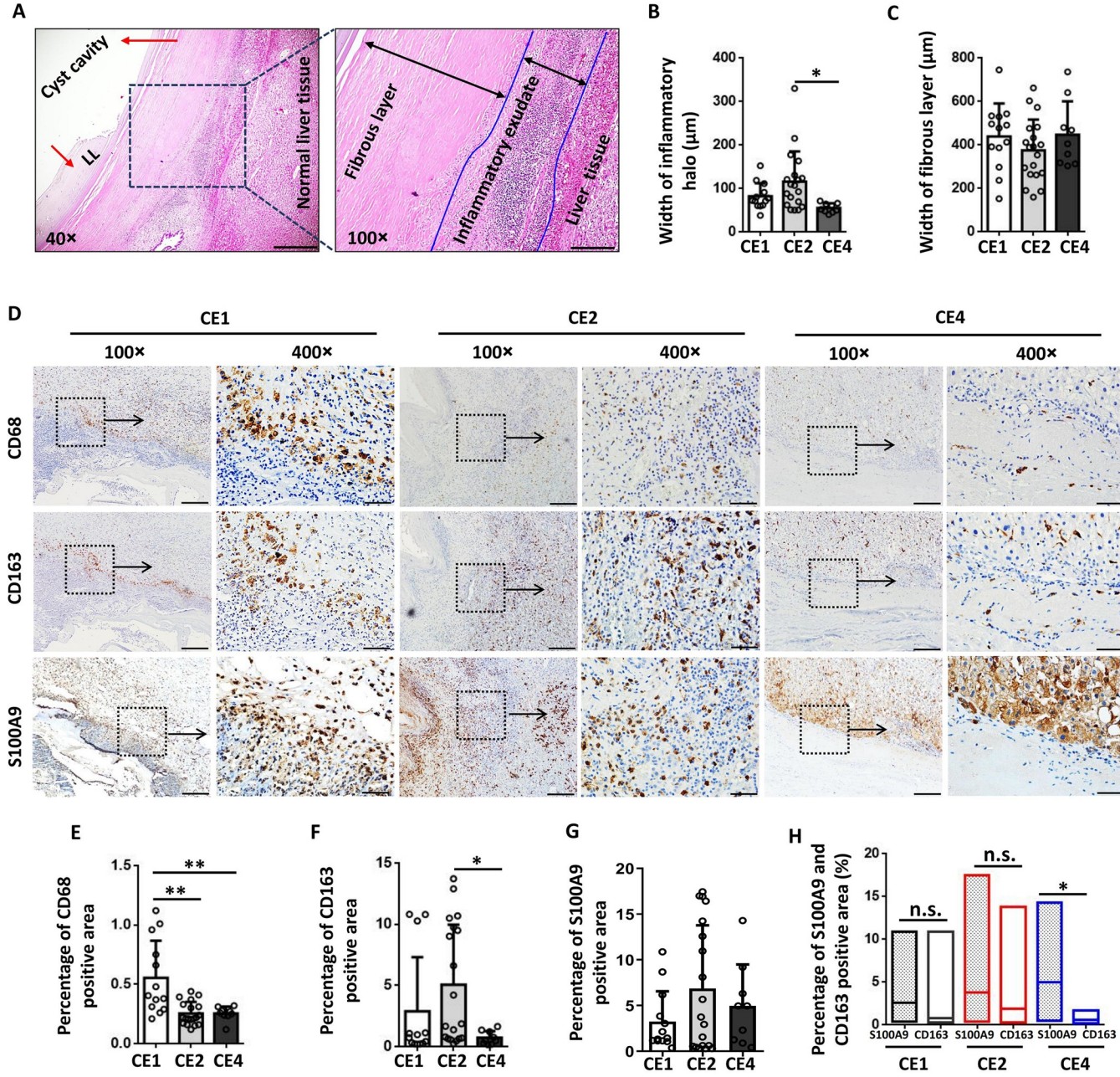

**Fig 1. Histopathological and immunohistochemical analyses of macrophage accumulation in livers from CE patients with different cyst activity stages.**
**(A)** Representative H&E staining of the cyst wall in liver tissue from CE patients. The cyst wall shows a laminated membrane (red arrow in the 40× image), a thick fibrous membrane (double-headed black arrow in the 100× image), and a peripheral halo of inflammatory exudate (delimited by medium-thickness blue lines indicated by the two-headed black arrow in the 100× image). The liver tissue adjacent to the halo of inflammatory exudate appears normal, as the liver tissue is distant from the cyst wall. **(B,C)** Thickness (μm) of the inflammatory halo around the cysts and the fibrous layer from CE patients with different cyst activity stages. **(D)** Representative immunohistochemical staining for CD68, CD163 and S100A9 in liver tissue samples from CE patients with different cyst activity stages (100× and 400× indicate the magnification of the figures; the bars represent 200 and 50 μm for the 100× and 400× images, respectively). **(E-G)** The percentages of positive CD68, CD163 and S100A9 staining were quantified using cellSens Dimension software. **(H)** Comparison of the S100A9-positive area percentage to the CD163-positive area percentage in CE patients with different cyst activity stages. (The results are presented as the means ± SEMs (n = 13, 19 and 9 for stages CE1, CE2 and CE4, respectively). *$P < 0.05$, **$P < 0.01$.

hemorrhagic lesions) on the liver surface on the second day after infection; however, from 5 days to 2 weeks post infection, the lesions gradually healed, and tiny yellowish-gray, opaque punctate lesions less than 1 mm in diameter were then observed. At 12 and 24 weeks, many small transparent hydatid cysts were distributed primarily at the edges of the liver lobes (**Fig 2A**).

H&E and IHC staining of *E. granulosus s.s.*-infected mouse livers showed that F4/80$^+$ macrophages infiltrated into the peripheral areas of the infectious foci at the early stage, markedly increasing from day 5 and peaking on day 8, accompanying the gradual infiltration of inflammatory cells around the infectious foci. Few larvae could be observed in the liver lobes at the early stage. At weeks 12 and 24, typical cystic structures with a clear GL and LL were observed; however, the abundance of aggregated inflammatory cells, including F4/80$^+$ macrophages, was significantly reduced in the pericystic area (percentage of F4/80-positive area, 2 days vs. 5 days vs. 8 days vs. 11 days vs. 2 weeks vs. 12 weeks vs. 24 weeks: 2.46 ± 0.36 vs. 22.36 ± 3.19 vs. 25.36 ± 2.21 vs. 20.39 ± 0.41 vs. 14.74 ± 1.42 vs. 5.34 ± 0.51 vs. 3.12 ± 0.58%, $P < 0.0001$) (**Fig 2B–2D**). In addition, α-SMA-positive cells and collagen (SR staining) accumulated around the infectious foci at the early stage, with gradually decreasing abundances at the late stage, consistent with F4/80$^+$ macrophage infiltration in the liver at different times after infection ($P < 0.0001$ for α-SMA-positive cells; $P < 0.0001$ for collagen) (**Fig 2C, 2E and 2F**).

## Hepatic macrophage composition changes in the mouse model during *E. granulosus s.s.* larval establishment

Hepatic macrophages, including resident KCs (CD45$^+$Ly-6G$^-$NK1.1$^-$CD19$^-$CD3$^-$CD11b$^{int}$F4/80$^{hi}$) and MoMFs (CD45$^+$Ly-6G$^-$NK1.1$^-$CD19$^-$CD3$^-$CD11b$^{hi}$F4/80$^{int}$), were evaluated in mice. Specifically, dynamic changes in the hepatic macrophage composition in NPLCs of mice infected with different *E. granulosus s.s.* PSC inoculum doses (LD, MD, and HD groups) at different times after infection were evaluated by flow cytometry. At 2 weeks, the percentage of KCs was significantly decreased in the MD and HD groups compared to the control group (Con vs. LD vs. MD vs. HD group: 31.04 ± 2.79 vs. 21.60 ± 1.90 vs. 15.74 ± 2.39 vs. 11.19 ± 1.02%, $P < 0.0001$), whereas the absolute numbers of KCs were not significantly different among the control, LD, MD and HD groups. In contrast, the percentage of MoMFs (Con vs. LD vs. MD vs. HD group: 6.09 ± 0.47 vs. 8.25 ± 0.58 vs. 23.08 ± 1.85 vs. 28.62 ± 3.24%, $P < 0.0001$) and absolute number of MoMFs ($P = 0.0006$) were significantly increased in the MD and HD groups and were even higher than those of KCs in the MD and HD groups (**Figs 3A–3C and S2A–S2B**). At 12 and 24 weeks, there were no significant differences in the percentages of KCs and MoMFs among the control, LD, MD and HD groups, but the percentages and absolute numbers of KCs were higher than those of MoMFs in the MD and HD groups. Moreover, the absolute numbers of KCs increased gradually in the MD and HD groups and were significantly higher in the HD group than in the control group at 12 weeks ($P = 0.0124$), whereas the absolute numbers of MoMFs decreased gradually in the MD and HD groups during the course of *E. granulosus s.s.* infection (**S2A and S2B Fig**).

## Hepatic macrophage phenotypic changes in the mouse model during *E. granulosus s.s.* larval establishment

To characterize phenotypic changes in mouse hepatic macrophages during *E. granulosus s.s.* infection, the expression of inducible nitric oxide synthase (iNOS; a proinflammatory phenotype marker) and CD206 (an anti-inflammatory phenotype marker) was evaluated in hepatic KC and MoMF subsets. In the proinflammatory phenotype analysis, at 2 weeks, the percentages of iNOS$^+$ KCs (Con vs. LD vs. MD vs. HD group: 0.82 ± 0.16 vs. 0.79 ± 0.15 vs.

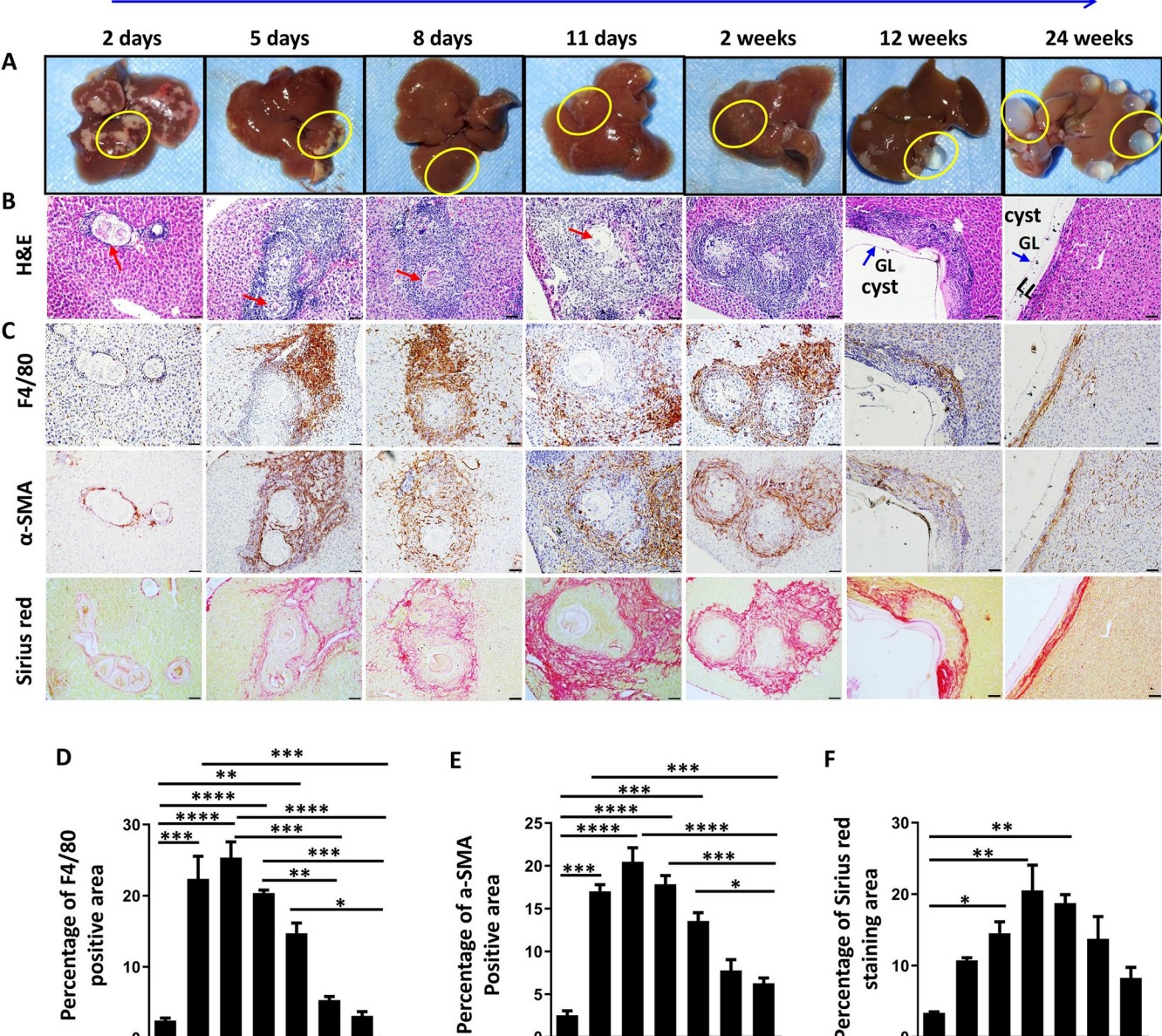

**Fig 2. Histopathological and immunohistochemical analyses of macrophage accumulation in mouse livers inoculated with *E. granulosus s.s.* PSCs at different times after infection.** (A) Representative macroscopic views of whole livers in mice infected with PSC inocula at different times after infection. The yellow circles highlight the lesions in the liver. The diffuse white patches indicate areas of necrosis, and the white spots indicate established parasites. (B) Representative H&E staining of infected livers. The larvae can be observed at the early stage (from 2 days to 2 weeks, red arrow). The cyst wall shows a clear GL, and an LL was observed at the late stage (12 and 24 weeks, blue arrow). GL: germinal layer; LL: laminated layer. (C) Representative immunohistochemical staining of F4/80 and α-SMA and SR staining in liver sections from *E. granulosus s.s.*-infected mice (200× indicates the magnification of the figures; the bars represent 100 μm). (D-F) The percentages of positive F4/80, α-SMA and SR staining areas in liver sections from 4–5 mice per time point. The data are shown as the means ±SEMs. *$P < 0.05$, **$P < 0.01$, ***$P < 0.001$, ****$P < 0.0001$.

5.25 ± 1.49 vs. 3.64 ± 0.72%, $P < 0.0001$) and iNOS⁺ MoMFs (Con vs. LD vs. MD vs. HD group: 9.74 ± 1.69 vs. 50.60 ± 8.95 vs. 51.80 ± 5.85 vs. 59.64 ± 2.18%, $P < 0.0001$) were significantly higher in the MD and HD groups than in the control group (**Figs 4A, 4B and S3**). The absolute numbers of iNOS⁺ MoMFs were significantly higher in the MD and HD groups than

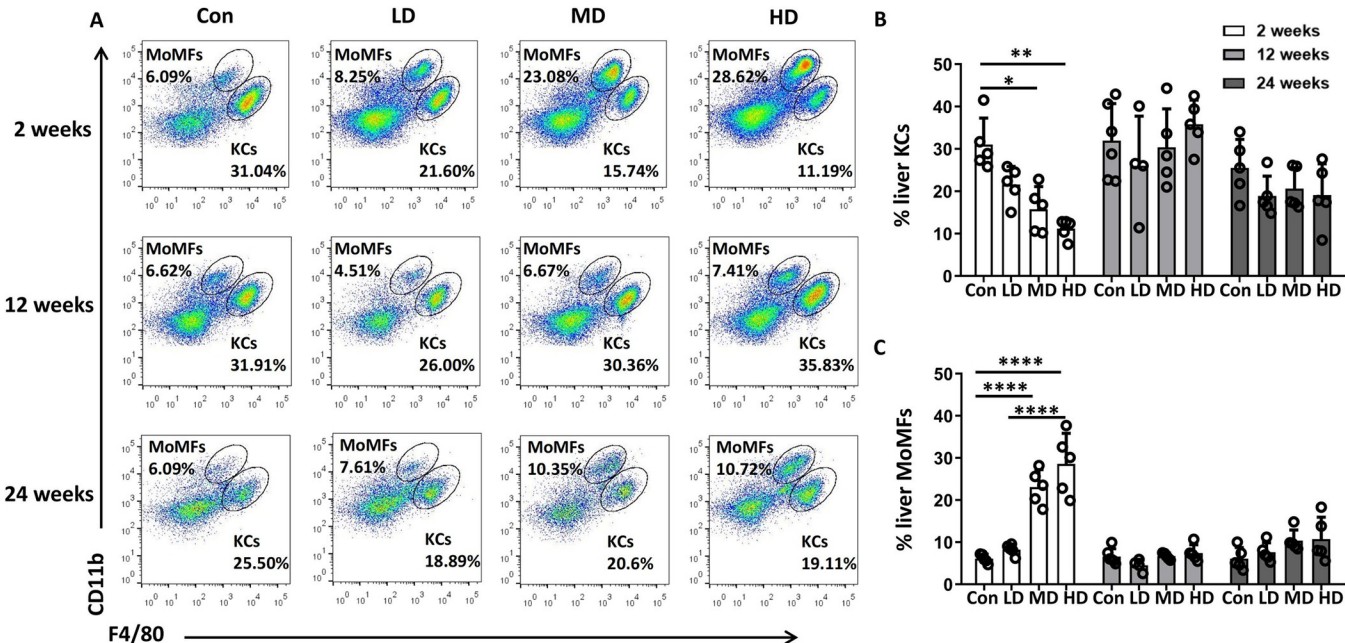

**Fig 3. Hepatic macrophage composition in mice infected with different inoculum doses of *E. granulosus s.s.* PSCs, evaluated in NPLCs, at different times after infection.** (A) Representative flow cytometry plots of hepatic macrophage subsets (KCs: CD11b$^{int}$F4/80$^{hi}$; MoMFs: CD11b$^{hi}$F4/80$^{int}$) detected among NPLCs from mice infected with different PSC inoculum doses at 2, 12 and 24 weeks after infection. (B, C) Percentages of KCs and MoMFs detected among NPLCs at 2, 12 and 24 weeks after infection. Con; LD: 50 PSCs; MD: 500 PSCs; HD: 2000 PSCs; n = 5–6 mice per group. The data are shown as the means ± SEMs. *$P < 0.05$, **$P < 0.01$ and ***$P < 0.001$.

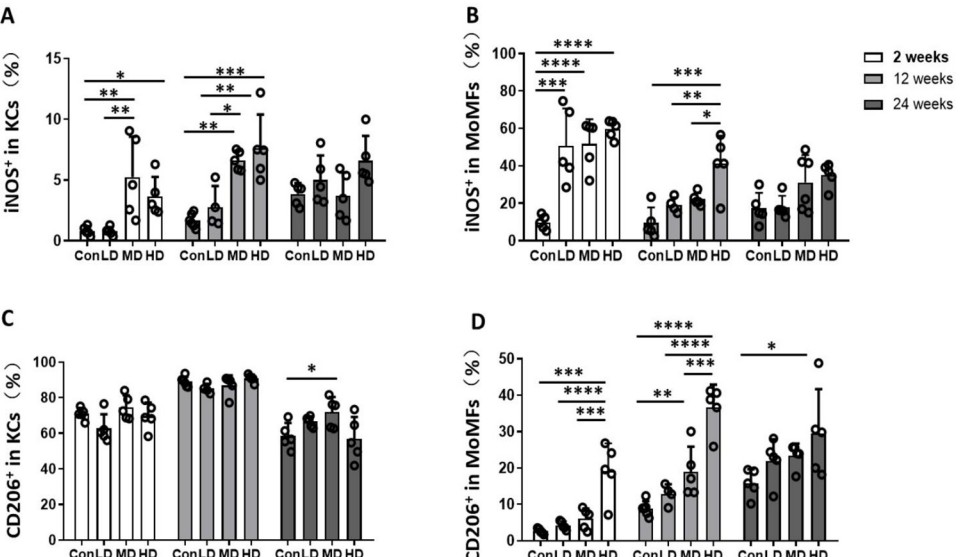

**Fig 4. Hepatic macrophage polarization phenotype in mice infected with different *E. granulosus s.s.* PSC inoculum doses, evaluated in NPLCs, at different times after infection.** (A, B) The percentages of iNOS$^+$ KCs and iNOS$^+$ MoMFs in the livers of mice infected with different PSC inoculum doses at 2, 12 and 24 weeks after infection. (C, D) The percentages of CD206$^+$ KCs and CD206$^+$ MoMFs in the livers of mice infected with different PSC inoculum doses at 2, 12 and 24 weeks after infection. Con; LD: 50 PSCs; MD: 500 PSCs; HD: 2000 PSCs; n = 5–6 mice per group. The data are shown as the means ± SEMs. *$P < 0.05$, **$P < 0.01$, ***$P < 0.001$ and ****$P < 0.0001$.

in the control group ($P < 0.0001$). At 12 weeks, the percentages and absolute numbers of iNOS+ KCs ($P = 0.0001$ for percentages; $P < 0.0001$ for absolute numbers) and iNOS+ MoMFs ($P = 0.0003$; $P = 0.0019$) were still significantly higher in the MD and HD groups than in the control group, but the absolute number of iNOS+ MoMFs in the HD group was significantly decreased compared to that at 2 weeks ($P = 0.0148$). However, at 24 weeks, the percentages and absolute numbers of iNOS+ KCs and iNOS+ MoMFs did not differ significantly among the control, LD, MD and HD groups (**Figs 4A, 4B, and S2C–S2D, and S3**).

In the anti-inflammatory phenotype analysis, the percentage of CD206+ KCs was not significantly different between these groups at 2, 12 and 24 weeks, but the percentage of CD206+ KCs (approximately 70% of the total KC population) was higher than that of iNOS+ KCs in the liver during the time course (**Figs 4C, S2E and S3**). In contrast, the percentage of CD206+ MoMFs was significantly higher in the HD group than in the control group at 2, 12 and 24 weeks ($P = 0.0012$ at 2 weeks; $P < 0.0001$ at 12 weeks; $P = 0.0436$ at 24 weeks). In addition, the absolute number of CD206+ MoMFs was significantly higher in the HD group than in the control group at 2, 12 and 24 weeks ($P < 0.0001$, $P = 0.0008$, $P = 0.0111$) (**Figs 4D, S2F and S3**).

## Depletion of hepatic macrophages promotes *E. granulosus s.s.* establishment and cyst growth by inhibiting CD4+ T-cell infiltration and liver fibrosis in a mouse model

To investigate the role of macrophages in the establishment and growth of *E. granulosus s.s.* hydatid cysts in a mouse model, we used CL to deplete hepatic macrophages in the mouse model at the early (1 and 2 weeks) and late stages (18–24 weeks) (**Fig 5A and 5B**). Higher numbers of infectious foci with PSCs were found in the livers of CL-treated mice compared with PL-treated control mice at 1 and 2 weeks ($P = 0.0159$) post infection (**Fig 5C and 5D**). The diameter (mm) of hydatid cysts in the liver was significantly increased in CL-treated mice at 24 weeks (PL vs. CL: $3.38 \pm 0.19$ vs. $7.54 \pm 0.48$ mm, $P = 0.0002$) (**Fig 5D and 5G**). Immunohistochemical staining showed that the populations of F4/80+ macrophages and CD4+ T cells were markedly decreased in the peripheral areas of the infectious foci in the livers of CL-treated mice compared with PL-treated control mice at 1 week (PL vs. CL, percentage of F4/80-positive area: $15.43 \pm 0.13$ vs. $1.64 \pm 0.51\%$, $P < 0.0001$; percentage of CD4-positive area: $6.01 \pm 0.47$ vs. $1.36 \pm 0.22\%$, $P = 0.0009$), 2 weeks (F4/80: $12.45 \pm 0.55$ vs. $2.28 \pm 0.16\%$, $P < 0.0001$; CD4: $5.36 \pm 0.70$ vs. $1.48 \pm 0.13\%$, $P = 0.0006$) and 24 weeks (F4/80: $4.17 \pm 0.64$ vs. $0.77 \pm 0.20\%$, $P = 0.0023$; CD4: $4.96 \pm 1.24$ vs. $1.47 \pm 0.21\%$, $P = 0.0327$) (**Figs 5C, 5E and S4**). In addition, hepatic macrophage depletion apparently attenuated liver fibrosis, as evaluated by SR and Col3α1 staining, at 1, 2 and 24 weeks (SR staining: $P = 0.0057$ at 1 week; $P < 0.0001$ at 2 weeks; $P = 0.0111$ at 24 weeks; Col3α1 staining: $P = 0.1000$ at 1 week; $P = 0.0317$ at 2 weeks; $P = 0.0571$ at 24 weeks) (**Figs 5C, 5F and S4**).

## Discussion

Hydatid cysts are the causative agent of CE, and they can survive for long periods in host organs (primarily the liver) [25]. Persistent *E. granulosus s.s.* infection elicits a granulomatous tissue reaction and gradually results in the formation of an immune microenvironment characterized by the accumulation of cells of monocytic origin and lymphocytes [8, 26]. Previous studies have demonstrated that macrophages are critical for granuloma formation, inflammatory progression and liver fibrosis development during parasitic infection [27]. However, the mechanism by which macrophages effectively contribute to the immune response during *E. granulosus s.s.* infection remains incompletely elucidated. Therefore, we focused on studying

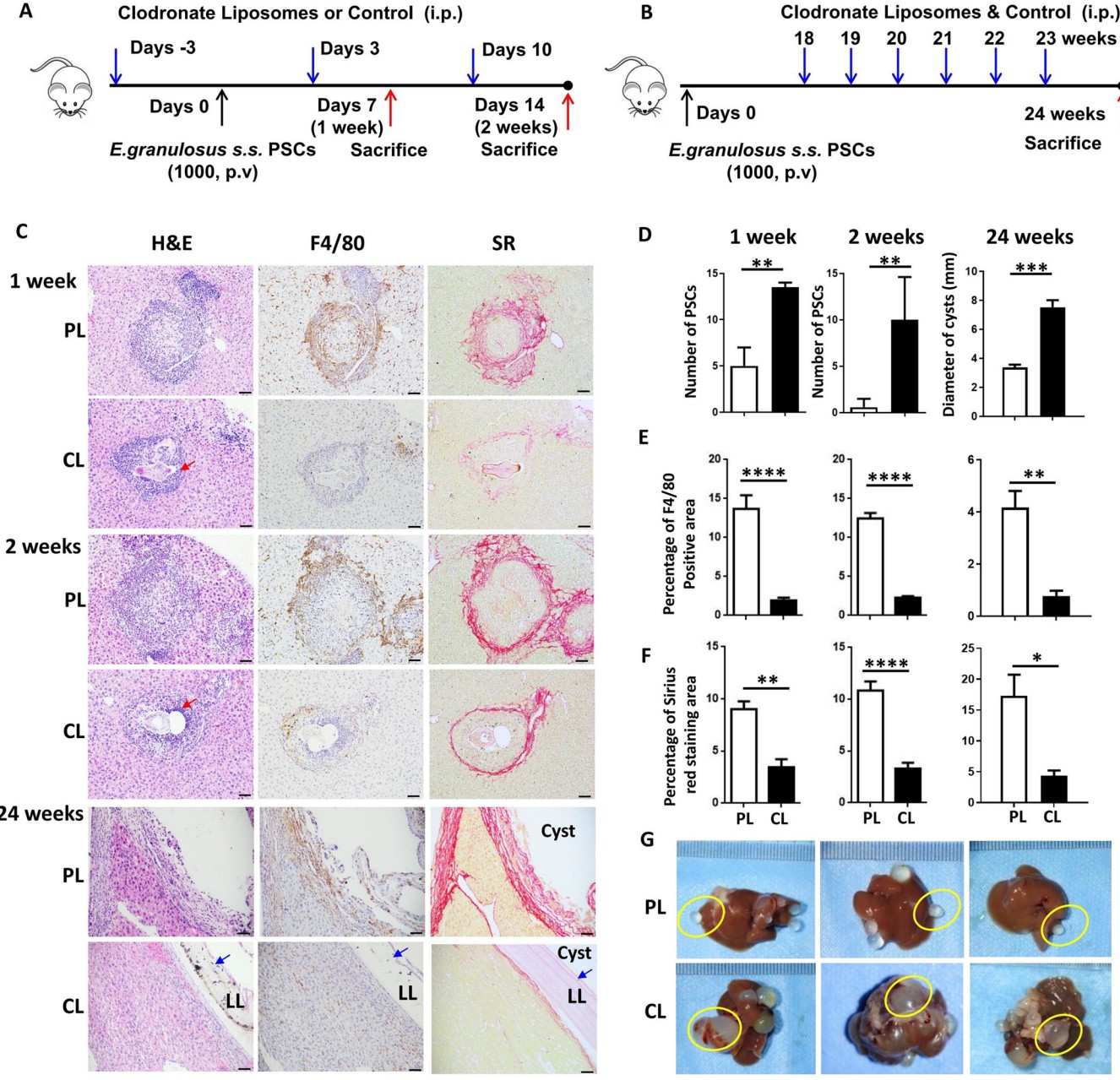

**Fig 5. Hepatic macrophage depletion in a mouse model promotes *E. granulosus s.s.* PSC establishment and cyst growth. (A, B)** Protocols for early and late hepatic macrophage depletion in the *E. granulosus s.s.* infection mouse model. **(C)** Representative H&E staining, immunohistochemical staining of F4/80, and SR staining in liver sections of mice treated with PL or CL at 1 and 2 weeks after inoculation (early stage) and at 24 weeks after inoculation (late stage) (200× magnification; scale bar, 100 μm). The red arrows indicate larvae, and the blue arrows indicate the LL structure. **(D)** Numbers of larvae in the livers of mice treated with PL or CL at 1 and 2 weeks after inoculation and the diameter (mm) of cysts in the livers of mice treated with PL or CL at 24 weeks after inoculation. **(E, F)** The percentages of positive F4/80 and SR staining in liver sections from mice treated with CL or PL at 1, 2 and 24 weeks after inoculation. **(G)** Representative images of hydatid cysts in the livers of *E. granulosus s.s.*-infected mice treated with PL or CL at 24 weeks after inoculation. The hydatid cysts are enclosed in the yellow circle. LL: laminated layer. CL: clodronate liposomes. PL: phosphate-buffered saline control liposomes. The data are shown as the means ± SEMs (n = 4–5 mice per group). *$P < 0.05$, **$P < 0.01$, ***$P < 0.001$ and ****$P < 0.0001$; n.s., nonsignificant.

the characteristics and function of hepatic macrophages during *E. granulosus s.s.* larval establishment and chronic infection.

The present study showed that inflammatory cells infiltrated around hydatid lesions to different degrees in CE patients. The width (μm) of the inflammatory halo was greater around active cysts (CE1 and CE2) than around inactive cysts (CE4), suggesting that hydatid cysts in the biologically active stage induce a stronger immune response in the hepatic lesion microenvironment. A previous study demonstrated that there is a correlation between cyst (PSC) viability and the overall level of inflammatory infiltration in the adventitial layer of fertile hydatid cysts; specifically, a low viability of fertile hydatid cysts correlates with a higher level of inflammatory infiltration [28]. In fact, the regulation of inflammation and/or immune responses is likely associated with strategies exploited by hydatid cysts to reprogram inflammatory cells, particularly local macrophages and dendritic cells, resulting in modulation of their ability to synthesize and/or release inflammatory mediators [29]. Consistent with previous studies [26], our data herein show that numerous CD68[+] macrophages were located within and around the sites of lymphocyte aggregation and that in almost all cases, these macrophages were lined up like a barrier just behind the fibrous capsule against the hydatid cyst (palisading macrophages). In addition, CD68[+] macrophages were more evident in the pericystic area in CE patients with active cysts, especially CE1 cysts, thus suggesting that the predominant infiltration of hepatic macrophages into the pericystic area was closely related to cyst activity (types CE1 and CE2) and might regulate disease progression [8].

Abundant evidence from an experimental mouse model has demonstrated that at the early stage, intraperitoneal larval infection can induce an intense inflammatory response that involves macrophages, lymphocytes and eosinophils [12, 30]. This peritoneal model is relatively easy to establish and safe but cannot reproduce the 'natural' site of initial parasite development, i.e., the liver [5]. The use of a primary infection model established by ingestion of infective eggs (as occurs in intermediate hosts) is more conducive to studying the host-parasite interaction; however, it is very risky for the operator to collect the parasite eggs from the definitive host (dog). In this study, we used C57BL/6 mice, a strain that properly mimics the host behavior observed in human infection (i.e., at the border of resistance and susceptibility), to establish a suitable hepatic experimental mouse model of *E. granulosus s.s.* by direct injection of precise numbers of PSCs via the portal vein, an approach that can reproduce the 'natural' liver location of initial parasite development [11, 31]. Our results showed that the number of F4/80[+] macrophages surrounding the PSC inoculation site was markedly increased beginning on day 5 and peaked on day 8 in the liver during the early stage and thereafter gradually decreased in the pericystic area from the early to the late stage, suggesting that the rapid increase in macrophages may be critical for regulating the early establishment and growth of the parasite in the lesion microenvironment. This phenomenon differs from our previous finding regarding *E. multilocularis* metacestodes, the causative agent of alveolar echinococcosis (AE), a similar infiltrative malignant tumor-like lesion, which induces a more intense inflammatory response and massive macrophage accumulation around the lesion at the late stage [4, 21]. In addition, the degree of F4/80[+] macrophage infiltration was consistent with the degree of liver fibrosis development during the course of infection, indicating that macrophages may be involved in liver fibrogenesis in the mouse model of *E. granulosus s.s.* infection.

Here, we found that although KCs respond to infection and seem to proliferate at low levels, the number of MoMFs was significantly increased and even exceeded that of resident KCs at the early stage of *E. granulosus s.s.* infection. This finding indicates that MoMFs are massively recruited to the infection site and thereby greatly expand the hepatic macrophage pool in the early infection stage. Furthermore, macrophages can polarize toward various phenotypes when stimulated by the surrounding microenvironment and parasitic infections [15, 27, 32,

33]. Some studies have reported that polarization of peritoneal macrophages toward an anti-inflammatory phenotype, thus enabling parasite survival, is also induced in the intraperitoneal infection model of *E. granulosus s.s.* infection [17, 34]. Our findings showed that infiltrated proinflammatory iNOS+ MoMFs (M1) accounted for the majority of the massive number of recruited macrophages in the lesion environment during the early stage of cyst establishment in the MD and HD groups; these cells may play an important role in aiding the clearance or killing of PSCs by releasing proinflammatory cytokines or nitric oxide (NO) via NOS-2/iNOS [35], the marker for M1 macrophages [36–38]. As the infection progressed, consistent with the events occurring during other parasitic diseases induced by *Schistosoma mansoni*, *Toxoplasma gondii* and *Clonorchis sinensis* [39–41], CD206+ KCs (M2) predominantly had an anti-inflammatory phenotype in the strongly Th2 cytokine-rich environment [42] during the late stage after infection, and a small number of recruited MoMFs gradually repolarized toward a predominant anti-inflammatory M2 phenotype, which promoted the survival and growth of established cyst forms in the liver in the murine model. Moreover, the percentage of CD163+ anti-inflammatory macrophages was obviously higher in patients with active CE than in patients with inactive CE, leading to chronic infection with *E. granulosus s.s.* Importantly, in patients with inactive CE4 cysts, macrophages exhibited a predominant proinflammatory S100A9+ M1 phenotype, which may limit hydatid cyst viability in the liver. Recent studies have also shown that some antigenic compounds (e.g., EgAgB and EgTPx) and the LL of the parasite contribute to driving macrophage differentiation toward an anti-inflammatory phenotype and then favor the establishment and growth of hydatid cysts, consistent with the Th2-regulated immune response in helminth infections [17, 43–45]. Furthermore, some studies have shown that tolerant B1 cells can polarize macrophages toward a more M2-like phenotype by secreting IL-10. Recently, the population of IL-10+ regulatory B (Breg) cells was reported to be increased at the late stage in an experimental model of *E. granulosus s.s.* infection [46–49]. Therefore, whether the increased population of Breg cells is associated with a phenotypic change in macrophages from M1 to M2 in the liver in mouse models requires further investigation.

To determine the functional role of macrophages in *E. granulosus s.s.* infection, we used CL to deplete macrophages in a mouse model. Macrophage depletion impairs PSC clearance in the liver at the early stage and reduces liver fibrosis, resulting in increased parasite burdens at the late stage. These findings confirmed the contribution of macrophages to the host defense against *E. granulosus s.s.* infection and were consistent with other reports regarding *E. multilocularis*, *Heligmosomoides polygyrus* and *Nippostrongylus brasiliensis* [21, 37, 50]. Recently, some studies have shown that hepatic macrophages also activate adaptive immune responses and exacerbate fibrosis by acting as professional antigen-presenting cells and secreting chemokines and cytokines [40, 51]. Hepatic macrophages may play a significant role in processing and presenting *E. granulosus s.s.* antigens to other immune cells, such as T cells, which are also abundant in lesion sites [26, 52]. In addition, an *in vitro* study demonstrated that macrophage depletion abolished the ability of this population to respond to T-cell mitogens or to PSCs [52]. Further studies have confirmed that macrophage-derived chemokines (including CCL24) and cytokines (including IL-2 and IL-12) promote the recruitment and activation of T cells, eosinophils, neutrophils and the corresponding immune responses, in addition to increasing the cytotoxicity of NK cells in the defense against parasitic [53, 54] or viral infections [55–59]. The present study shows that macrophage depletion reduces the accumulation of CD4+ T cells around hepatic lesions, possibly due to inhibition of CD4+ T-cell recruitment during *E. granulosus s.s.* infection. Moreover, recent studies by our group and others have shown that CD4+ T cells are involved in controlling the parasite burden and that CD4+ T-cell deficiency promotes the development and growth of hydatid cysts during infection [8, 10]. Taken together, these data indicate that hydatid cysts may evolve to exploit macrophages as

their main target cells and suppress the direct and indirect antiparasitic effects exerted predominantly by T cells by regulating macrophage polarization, thus promoting *E. granulosus s. s.* infection in a favorable immune microenvironment. This possibility that needs full confirmation.

In conclusion, our study demonstrates that hepatic macrophages are involved in the formation of the immune microenvironment in the liver in both CE patients and *E. granulosus s.s.*-infected mice. During the early stage of infection, hepatic macrophages predominantly have a proinflammatory phenotype that promotes parasite clearance. Subsequently, the environment during chronic infection drives hepatic macrophage polarization toward an anti-inflammatory phenotype that favors persistent infection. In addition, the phenotype of macrophages is shaped by the infectious dose and the progression of *E. granulosus s.s.* PSC infection in the liver. Importantly, the data from our mouse model indicated that depletion of hepatic macrophages promotes *E. granulosus s.s.* establishment and growth partially by inhibiting liver fibrosis and CD4+ T-cell recruitment. This study suggests that hepatic macrophages play a central role in the pathogenesis of CE and that exploring strategies to target hepatic macrophages may help to identify novel therapeutic approaches for CE.

## Supporting information

**S1 Data. Excel spreadsheet containing, in separate sheets, the raw data for Figs 1, 2, 3, 4, 5, S1, S2, S3, and S4.**
(XLS)

**S1 Table. Demographical and clinical characteristics of the CE patients.**
(DOCX)

**S1 Fig. Analysis of liver fibrosis in liver tissue samples from cystic echinococcosis (CE) patients with different cyst activity stages. (A)** Representative degrees of liver fibrosis, as determined by α-SMA, SR and Masson staining, in liver tissue sections from CE patients with different cyst activity stages (100× and 400× indicate the magnification of the figures; the bars represent 200 and 50 μm for 100× and 400×, respectively). **(B-D)** The percentages of positive α-SMA, SR and Masson staining were quantified using cellSens Dimension software. The results are presented as the means ± SEMs (n = 13, 19 and 9 for stages CE1, CE2 and CE4, respectively).
(TIF)

**S2 Fig. Absolute quantification of hepatic macrophage subsets in mice infected with different *E. granulosus s.s.* PSCs) inoculum doses, evaluated in NPLCs, at different times after infection. (A, B)** Absolute numbers of KCs and MoMFs among the NPLCs of mice infected with different PSC inoculum doses at 2, 12 and 24 weeks after infection. **(C, D)** Absolute numbers of iNOS+ KCs and iNOS+ MoMFs among the NPLCs of mice infected with different PSC inoculum doses at 2, 12 and 24 weeks after infection. **(E, F)** Absolute numbers of CD206+ KCs and CD206+ MoMFs among the NPLCs of mice infected with different PSC inoculum doses at 2, 12 and 24 weeks after infection. Con; LD: 50 PSCs; MD: 500 PSCs; HD: 2000 PSCs; n = 5–6 mice per group. The data are shown as the means ± SEMs, *$P < 0.05$, **$P < 0.01$, ***$P < 0.001$ and ****$P < 0.0001$.
(TIF)

**S3 Fig. Representative flow cytometry plots of the macrophage polarization phenotype gated on liver KC and MoMF subsets in mice infected with different *E. granulosus s.s.* PSCs) inoculum doses, evaluated in NPLCs, at different times after infection. (A)**

Intracellular staining of iNOS in KCs and MoMFs in the livers of mice infected with different PSC inoculum doses at 2, 12 and 24 weeks post infection. **(B)** Intracellular staining of CD206 KCs and MoMFs in the livers of mice infected with different PSC inoculum doses at 2, 12 and 24 weeks post infection. Con; LD: 50 PSCs; MD: 500 PSCs; HD: 2000 PSCs.
(TIF)

**S4 Fig. Hepatic macrophage depletion reduces Col3α1 expression and CD4$^+$ T-cell accumulation in the inflammatory cell zone around liver lesions in *E. granulosus s.s.*-infected mice. (A)** Representative immunohistochemical staining of Col3α1 in liver sections from mice treated with PL or CL at 1, 2 and 24 weeks after inoculation (200× magnification; scale bar, 100 μm). **(B)** The percentages of positive Col3α1 staining in the liver sections. **(C)** Representative immunohistochemical staining of CD4 in liver sections from mice treated with PL or CL at 1, 2 and 24 weeks after inoculation (200× magnification; scale bar, 100 μm). **(D)** The percentages of positive CD4 staining in the liver sections. The red arrows indicate the larvae and LL structure. LL: laminated layer. CL: clodronate liposomes. PL: phosphate-buffered saline control liposomes. The data are shown as the means ± SEMs (n = 4–5 mice per group).
$^*P < 0.05$, $^{**}P < 0.01$ and $^{***}P < 0.001$.
(TIF)

## Acknowledgments

We thank the patients who voluntarily participated in the study and the animal experiment center of Xinjiang Medical University.

## Author Contributions

**Conceptualization:** Hui Wang, Chuanshan Zhang.

**Data curation:** Hui Wang.

**Formal analysis:** Hui Wang, Qian Yu, Mingkun Wang.

**Funding acquisition:** Hui Wang, Chuanshan Zhang.

**Investigation:** Jiao Hou, Maolin Wang.

**Methodology:** Hui Wang, Qian Yu, Mingkun Wang, Xinling Hou, Zibigu Rousu.

**Project administration:** Hui Wang.

**Resources:** Jiao Hou, Maolin Wang, Tiemin Jiang, Hao Wen.

**Software:** Hui Wang, Mingkun Wang, Jiao Hou, Dewei Li.

**Supervision:** Hui Wang, Chuanshan Zhang.

**Visualization:** Qian Yu, Xuejiao Kang, Jing Li.

**Writing – original draft:** Hui Wang, Qian Yu, Chuanshan Zhang.

**Writing – review & editing:** Hui Wang, Hao Wen, Chuanshan Zhang.

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
