## [Decision Letter · Decision Letter 0]

8 Aug 2023

Dear Dr. Zhang,

Thank you very much for submitting your manuscript "Hepatic macrophages play critical roles in establishment and growth of hydatid cysts in the liver during Echinococcus granulosus sensu stricto infection" for consideration at PLOS Neglected Tropical Diseases. As with all papers reviewed by the journal, your manuscript was reviewed by members of the editorial board and by several independent reviewers. In light of the reviews (below this email), we would like to invite the resubmission of a significantly-revised version that takes into account the reviewers' comments. 

The reviewers have pointed out several key elements of the manuscript that require attention, including but not limited to the results of the analysis of human infections. In addition to the comments of the reviewers, please also include a discussion of the limitations of the murine model, as it is a secondary infection model (using protoscoleces), and the early time points may not be representative of primary infections produced by infective eggs (as occurs in human patients).

We cannot make any decision about publication until we have seen the revised manuscript and your response to the reviewers' comments. Your revised manuscript is also likely to be sent to reviewers for further evaluation.

Sincerely,

Delia Goletti, M.D., Ph.D.

Academic Editor

Uriel Koziol

Section Editor

Reviewer's Responses to Questions

**Key Review Criteria Required for Acceptance?**

**Methods**

-Are the objectives of the study clearly articulated with a clear testable hypothesis stated?

-Is the study design appropriate to address the stated objectives?

-Is the population clearly described and appropriate for the hypothesis being tested?

-Is the sample size sufficient to ensure adequate power to address the hypothesis being tested?

-Were correct statistical analysis used to support conclusions?

-Are there concerns about ethical or regulatory requirements being met?

Reviewer #1: (No Response)

Reviewer #2: (No Response)

**Results**

-Does the analysis presented match the analysis plan?

-Are the results clearly and completely presented?

-Are the figures (Tables, Images) of sufficient quality for clarity?

Reviewer #1: (No Response)

Reviewer #2: (No Response)

**Conclusions**

-Are the conclusions supported by the data presented?

-Are the limitations of analysis clearly described?

-Do the authors discuss how these data can be helpful to advance our understanding of the topic under study?

-Is public health relevance addressed?

Reviewer #1: (No Response)

Reviewer #2: (No Response)

**Editorial and Data Presentation Modifications?**

Reviewer #1: (No Response)

Reviewer #2: (No Response)

**Summary and General Comments**

Reviewer #1: (No Response)

Reviewer #2: In this study, the authors characterize the immune events involved in the establishment and growth of echinococcal cysts focusing on the role of macrophages.

The in vivo studies, in which timely evaluation is performed, are properly designed and allow to follow the echinococcal cyst from the infection to its establishment and growth, and therefore to characterize the role of macrophages in the different stages of infection.

In contrast, the characterization of the “history” of echinococcal cysts in humans is a complex topic. Indeed, the current knowledge on the natural history of the cysts rely on mass screening studies by ultrasound; moreover, we need to consider that cysts with morphological aspects suggesting viability may actually be not viable and vice versa. Based on this assumption, the main concern regarding this study is the lack of study population description for the human study. A paragraph in the RESULTS section on the study population is missing. Inclusion and exclusion criteria must be reported; moreover, for each clinical group it is important to know if enrolled patients have only hepatic cysts or have cysts located in both liver and other organs; data on pharmacological treatment prior enrollment are also important. Pharmacological treatment information are crucial for clinical stage definition mainly for CE4 cysts as, CE4 cysts resulting from treated CE3b, even presenting pathognomonic features of inactive cysts, may reactivate overtime. Therefore, CE4 cysts may be considered inactive and not viable only if spontaneously inactivated or if the last treatment goes back several years before enrollment. The authors refer to the WHO-IWGE classification system for CE1-CE5 staging, however, imprecise concepts are reported. Indeed, the classification in active/transitional/inactive clinical groups is reported in ref 19 (WHO Informal Group Acta Trop 2003), whereas ref 20 (Hosch et al 2008) correlates the metabolite profiles of cysts with cyst activity/viability defined by ultrasound. Moreover, Hosch et al reports that CE3a have equal probability of being viable or nonviable, whereas CE3b cysts are usually viable (in the manuscript CE3 are described as CE3a!!!). Finally, a Table reporting all patients demographical and clinical characteristics is needed.

Minor points: 

1. English and editing need to be deeply revised

2. Lines 115-117: Please revise the sentence: the “early” and the “late” reaction to E.granulosus in humans is difficult to define.

3. Line 206-207: “cells were first….at 4°C and” is redundant

4. In the MATERIAL and METHODS section the authors stated that 9 patients with CE4 or CE5 cysts were enrolled. However, only data on CE4 patients are reported. Why patients with CE5 were excluded? If CE4-CE5 patients are considered as a single group, this should be clearly stated in the RESULTS section and figures should be modified accordingly. Moreover, please consider the importance of treatment for groups classification.

5. In the flow cytometry paragraph of the MATERIAL and METHODS section: how many cells per group were acquired at 2, 12 and 24 weeks?

6. lines 257-262: the results should be described more precisely: CD68 is significantly increased in CE1 compared to CE2 (that is a biologically viable stage) and to CE4. Moreover, CE2 and CE4 have similar percentages of CD68+ cells. Could the author comment these data?

7. What the author means for “absolute number” of KCs or MoMFs? In the MATERIAL and METHODS section only percentage of positive cells is mentioned.

8. Lines 318-330: the results should be described more precisely.

PLOS authors have the option to publish the peer review history of their article (what does this mean?). If published, this will include your full peer review and any attached files.

Reviewer #1: No

Reviewer #2: No
---

## [Editor Report · Decision Letter 1]

21 Oct 2023

Dear dr Chuanshan Zhang,

We are pleased to inform you that your manuscript 'Hepatic macrophages play critical roles in the establishment and growth of hydatid cysts in the liver during Echinococcus granulosus sensu stricto infection' has been provisionally accepted for publication in PLOS Neglected Tropical Diseases.

Best regards,

Delia Goletti, M.D., Ph.D.

Academic Editor

Uriel Koziol

Section Editor

---

## [Editor Report · Acceptance letter]

29 Oct 2023

Dear Professor Zhang,

We are delighted to inform you that your manuscript, " Hepatic macrophages play critical roles in the establishment and growth of hydatid cysts in the liver during *Echinococcus granulosus* sensu stricto infection ," has been formally accepted for publication in PLOS Neglected Tropical Diseases.

Best regards,

Shaden Kamhawi

co-Editor-in-Chief

Paul Brindley

co-Editor-in-Chief
